# One Question at a Time: A Semantic Bottleneck for Interpretable Visual Brain Decoding from fMRI

**Sara Cammarota**\*
Department of Biomedicine and Prevention
University of Rome, Tor Vergata
Viale Montpellier, 1 – Rome (IT)
`sara.cammarota@uniroma2.eu`

**Matteo Ferrante**\*
Tether Evo
Department of Biomedicine and Prevention
University of Rome, Tor Vergata
`matteo.ferrante@tether.io`

**Nicola Toschi**
Department of Biomedicine and Prevention
University of Rome, Tor Vergata
Martinos Center For Biomedical Imaging
MGH and Harvard Medical School (USA)
`toschi@med.uniroma2.it`

## Abstract

Decoding of visual stimuli from noninvasive neuroimaging techniques such as functional magnetic resonance (fMRI) has advanced rapidly in the last few years; yet, most high-performing brain decoding models rely on complicated, non-interpretable latent spaces (e.g., CLIP). In this study, we present an interpretable brain decoding framework; our key innovation is the insertion of a semantic bottleneck into BrainDiffuser, a well established brain decoding pipeline. Specifically, we build a $214 -$ dimensional binary interpretable space $\mathcal{L}$ for images, in which each dimension answers a specific question about the image (e.g., "Is there a person?", "Is it outdoors?"). A first ridge regression maps the activity of the voxel to this semantic space. Because this mapping is linear, its weight matrix can be visualized as maps of voxel importance for each dimension of $\mathcal{L}$, revealing which cortical regions most influence each semantic dimension. A second regression then transforms these concept vectors into CLIP embeddings required to produce the final decoded image, conditioning the BrainDiffuser model. We find that voxel-wise weight maps for individual questions are highly consistent with canonical category-selective regions in the visual cortex (face, bodies, places, words), simultaneously revealing that activation distributions, not merely location, bear semantic meaning in the brain. Visual brain decoding performances are only slightly lower compared to the original BrainDiffuser metrics (e.g., CLIP similarity is decreased by $\leq 4\%$), but offering substantial gains in interpretability and neuroscientific insights. These results show that our interpretable brain decoding pipeline enables voxel-level analysis of semantic representations in the human brain without sacrificing decoding accuracy.

## 1 Introduction

Decoding the content of the human mind from neural activity is one of the main challenges of contemporary neuroscience, aiming to classify, retrieve, or reconstruct the stimuli that elicited a

---

\*Equal contribution.

Preprint.

certain neural response, in an effort to translate the language of the human mind into a representation of the external world (visual scenes, linguistic input, music, or other experiential domains) [4, 8, 9, 12, 14, 23, 17, 19, 24, 25, 26, 33, 43, 37, 45]. Advances in AI and the availability of large, high-quality neuroimaging datasets [2, 7, 22, 27, 34, 28] have enabled the decoding of visual [40, 10, 42, 18, 20, 30, 21], language [3, 13, 43], semantic [15], and musical [14, 5, 16] content, even from noninvasive modalities such as fMRI.

For visual brain decoding specifically, several methods have converged on a dual-stream design, predicting high-level semantic features and low-level structural features, combined with a generative model for image reconstruction [31, 35, 40]. In particular, even simple linear regressions from brain activity to latent semantic spaces (such as those obtained by CLIP) achieve strong generalizable performance [36, 41], although these latent spaces remain difficult to interpret, as it is not clear what each dimension corresponds to. Here, we introduce an interpretable brain-decoding framework that bridges this gap. We propose an approach to map brain activity into a human-readable embedding space that serves as the middle ground between neural correlates and the semantic embeddings of CLIP. Each dimension of this intermediate space corresponds to a visual concept (such as *people* or *motion*), allowing us to gain interpretability in the input space. We insert this framework within BrainDiffuser[40], a well-established reconstruction pipeline, to test whether interpretability affects image reconstruction performance.

Our work draws inspiration from [6], which replaced opaque embeddings with interpretable vectors in language brain encoding, maintaining high accuracy, and from [32], which showed that bottlenecks in visual decoding pipelines can preserve performance, suggesting lower-dimensional brain representations. While most literature focuses on optimizes decoding accuracy using non-interpretable embeddings, we take a complementary approach: using high-performing models to probe the structure of conceptual representations in the brain.

The main contributions of this work are: (i) we introduce a semantic bottleneck, interpretable by design that preserves the information required for high-quality image reconstruction while providing each dimension with a clear meaning; (ii) we decompose the *brain to semantic* mapping into two separate maps: brain to interpretable space, and interpretable space to semantic CLIP space. This yields voxel-level insights that are easily visualized and compared to known activation patterns in the human brain; (iii) we show that the proposed model produces stable, anatomically plausible concept maps across subjects, in which the distribution of voxel-pattern strength, not only the voxels location, carries important information.

## 2    Methods

We used the Natural Scenes Dataset (NSD)[2], a deep fMRI data set that encompasses eight healthy adult subjects who performed a continuous recognition task on thousands of images from the COCO dataset. For further details on the data, please refer to Appendix 5.1.

To develop our interpretable pipeline, we build on a classical brain decoding method for images, BrainDiffuser[40]. We choose BrainDiffuser because of its strong performance, modular architecture, and simplicity of implementation. Its clarity and flexibility make it well-suited for integrating our interpretable semantic bottleneck without introducing additional confounds. The original BrainDiffuser can be described as a two-stage image reconstruction framework from fMRI signals. In stage 1, a pre-trained VDVAE[11] encoder maps images to hierarchical latents, of which the first 31 layers are linearly predicted from fMRI and decoded into a coarse image $64 \times 64$. Stage 2 refines this through Versatile Diffusion (VD)[46], conditioning on the CLIP-Vision and CLIP-Text embeddings predicted from fMRI, and initialized with the Stage 1 output. More details are provided in Appendix 5.2. In particular, in the second stage of the original BrainDiffuser brain activity $Z$ is mapped to the estimated CLIP embeddings by means of a linear model $\widetilde{\mathbf{h}} = ZW$ where $W$ has shape $(n_{\text{voxels}}, \text{CLIP}_{\text{dimension}})$. Because the CLIP embeddings are not interpretable, it is difficult to understand the meaning of the model weight matrix $W$.

To understand how brain activity is converted into the representations given by the CLIP embeddings, we factorize the decoding weight matrix $W$ in two separate matrices $A$ of size $n_{\text{voxels}} \times a$ and $B$ of size $a \times \text{CLIP}_{\text{dimension}}$ such that $W = AB$. The core idea is that we can design an $a-$ dimensional, semantically interpretable latent space $\mathcal{L}$ so that $A : \text{brain activity} \rightarrow \mathcal{L}$ and $B : \mathcal{L} \rightarrow \text{CLIP space}$. This decomposition allows for direct inspection of $A$, whose entries reveal the individual contribution

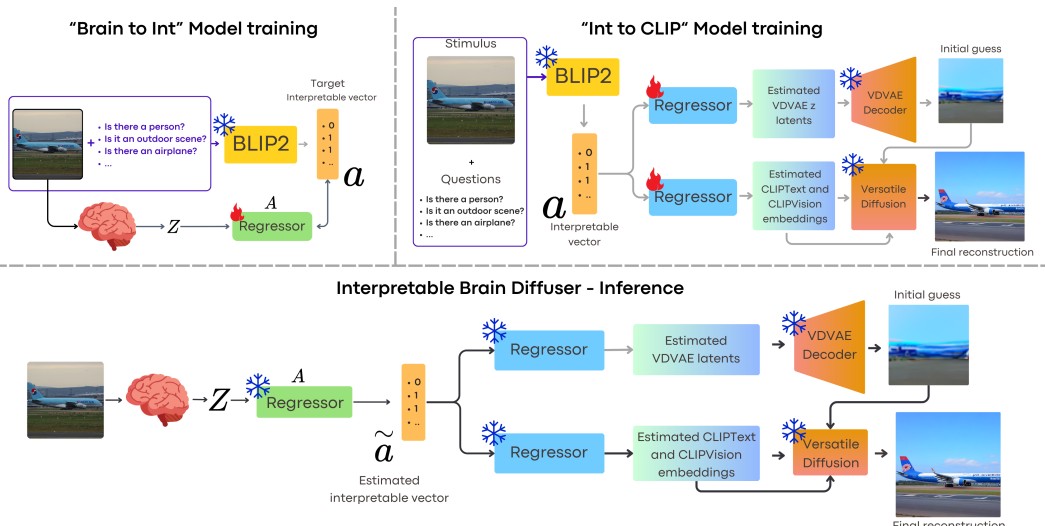

Figure 1: "Brain to Int" Model training: training pipeline of the model mapping from brain activity $\rightarrow \mathcal{L}$. "Int to CLIP" Model training: training pipeline of the model mapping from $\mathcal{L} \rightarrow$ CLIP/VDVAE space. The obtained VDVAE, CLIP embeddings are used to generate the reconstructed image. "Interpretable BrainDiffuser - Inference": the whole inference pipeline of our interpretable model. Neural data $Z$ is first mapped to the interpretable space $\mathcal{L}$ via the frozen "Brain to Int" model and then mapped to CLIP/VDVAE space via the frozen "Int to CLIP" model).

of each brain voxel to every latent dimension. Therefore, the factorization exposes a linear map onto semantic dimensions, enabling neuroscientific interpretation.

To design the latent space $\mathcal{L}$ we use a two-step process. First, GPT-4o [38] generates a set of non-overlapping binary questions describing NSD images (e.g., "Is there a person?", "Is it outdoors?"), based on training image captions. After removing duplicates, we obtain 214 questions, corresponding to $\mathcal{L}$'s dimensions. This size imposes a 20–30% bottleneck relative to CLIP's 768 dimensions, consistent with prior work showing that even 50 dimensions can preserve most performance [32]. The full question list is given in Appendix 5.8. Second, BLIP-2 [29] answers all these questions sequentially for each image, producing interpretable embeddings. We validate that $\mathcal{L}$ preserves CLIP-level semantics by aligning it to CLIP-Vision via ridge regression and measuring the top-$k$ retrieval accuracy (details in Appendix 5.5).

We first estimate the linear map $B : \mathcal{L} \rightarrow$ CLIP by duplicating the BrainDiffuser architecture and training a Ridge regression to reconstruct the images from ground-truth interpretable embeddings (instead of brain activity). This latent-to-image model verifies that the designed latent space is sufficiently informative, as high reconstruction quality indicates that the latent space encodes important properties of the NSD images. Moreover, this model is subsequently frozen and applied to the interpretable embeddings inferred from brain activity to generate the final reconstructed images. We then fit a ridge regression model to estimate the normalized coordinates of the interpretable space $\mathcal{L}$ from brain activity. The learned map $A$, one per subject, is interpretable by design: each $i-$th column encodes the contribution of every brain voxel to the $i-$th coordinate in $\mathcal{L}$, i.e. the answer to each $i-$th binary question. Figure 1 outlines the full pipeline of our model.

Visualization of the brain regions most involved in the estimation of each dimension of $\mathcal{L}$ was achieved by first mapping each Region Of Interest (ROI) to the actual anatomical space of each participant. We then register these coordinates to the MNI-152 standard space and compared the resulting maps to visual cortex regions knowingly related to concepts such as bodies, faces, words and places. For final visualizations, we kept only the top 4% of most influential voxels and discarded any cluster smaller than 100 voxels to suppress noise. The decision to focus on the top 4% of most influential voxels was made empirically, guided by two considerations. First, this threshold yields a voxel count that is comparable to that of well-characterized functional regions in the dataset, such as face-, body-, place-, and word-selective areas, each comprising a similar number of voxels[2]. Second, this proportion provides a clear and interpretable visualization.

# 3 Results

Table 1 presents the results of the interpretable model (Int-BD) evaluation compared to the original BrainDiffuser and to the intermediate model *Int to CLIP*. The latter serves as a check that the designed interpretable embeddings are sufficiently good for image reconstruction. Even if it obtains the best high-level performance across all models, this is not a brain decoding model; its superior accuracy stems from being the only model to bypass the noisy fMRI data, mapping directly the true interpretable embeddings to the CLIP space. Our interpretable model retains good reconstructing performance (e.g., on average $-4\%$ on CLIP score and $-8\%$ on 50-way-top-1 accuracy with respect to the original BrainDiffuser), confirming that interpretability does not affect excessively the reconstructing accuracies (see the Appendix 5.6 for examples of decoded and retrieved images from brain activity).

Table 1: Average quantitative analysis of reconstructed images with low- (top) and high- (bottom) level metrics (mean $\pm$ std across Subj01/02/05/07). ($\uparrow$): higher is better; ($\downarrow$): lower is better. See 5.4 for details on the used metrics.

| Model | PixCorr $\uparrow$ | SSIM $\uparrow$ | MSE $\downarrow$ | Cosine Sim. $\uparrow$ | AlexNet(2) $\uparrow$ | AlexNet(5) $\uparrow$ |
|---|---|---|---|---|---|---|
| BrainDiffuser | $0.238 \pm 0.034$ | $0.351 \pm 0.007$ | $0.128 \pm 0.007$ | $0.788 \pm 0.009$ | $0.923 \pm 0.021$ | $0.953 \pm 0.011$ |
| Int-BD | $0.150 \pm 0.005$ | $0.319 \pm 0.001$ | $0.141 \pm 0.001$ | $0.764 \pm 0.002$ | $0.791 \pm 0.007$ | $0.869 \pm 0.007$ |
| Int to CLIP | $0.156 \pm 0.002$ | $0.311 \pm 0.001$ | $0.149 \pm 0.001$ | $0.757 \pm 0.001$ | $0.814 \pm 0.005$ | $0.895 \pm 0.001$ |

| Model | IncepV3 $\uparrow$ | CLIP $\uparrow$ | EffNet Dist. $\downarrow$ | SwAV Dist. $\downarrow$ | 50-way-top-1 Acc. |
|---|---|---|---|---|---|
| BrainDiffuser | $0.907 \pm 0.014$ | $0.915 \pm 0.013$ | $0.718 \pm 0.015$ | $0.416 \pm 0.010$ | $0.525 \pm 0.021$ |
| Int-BD | $0.846 \pm 0.012$ | $0.877 \pm 0.009$ | $0.788 \pm 0.014$ | $0.475 \pm 0.008$ | $0.445 \pm 0.024$ |
| Int to CLIP | $0.941 \pm 0.005$ | $0.952 \pm 0.002$ | $0.663 \pm 0.006$ | $0.414 \pm 0.003$ | $0.688 \pm 0.010$ |

Figure 2 displays a subset of columns of the interpretable matrix $A$, mapped to the MNI-512 space, compared to the standard ROIs for canonical categories. The so-obtained activation maps exhibit a good similarity with standard concept-related regions, and the specific distribution of voxel activations within the same region varies with each sub-concept examined. For instance, the reference region for bodies displays different activation patterns depending on the specific question. Even though all activations tend to overlap with the reference regions, the distribution of voxel intensity varies depending on the specific subconcept. For example, in the body ROI, "Is someone running?" produces a broad low-level pattern with a few intense clusters, whereas "Is someone dancing?" yields stronger activation of those clusters with minimal surrounding activity. Similar comments can be made for place-related activations (e.g. the questions "Is the image taken outdoors?", "Is the image taken on a street?"). We don't observe a sharp distinction in the activations for faces and bodies. Neuroscientific studies have shown that face- and body-selective areas are anatomically close in the human (and monkey) high-level visual cortex and often co-activate when a whole person or animal is present [44]. This is simply explainable because, under normal circumstances, faces and bodies are seen together. For many of the questions related to bodies and faces we observed an overlap with both face and body regions. For example, the question "Is the main subject smiling?" shows strong overlap with face-selective regions but also notable overlap with body-related areas.

# 4 Discussion and Conclusions

We presented an interpretable fMRI-to-image decoding model that maintains high reconstruction accuracy while enabling voxel-level neuroscientific insight. A semantic bottleneck maps voxel signals to a human-readable concept space before projection into CLIP embeddings, linking low-level BOLD signals to semantic concepts. This design yields only minor drops in low- and high-level metrics compared to the noninterpretable BrainDiffuser baseline. This small degradation is expected: the prompts are derived from human-authored COCO captions via BLIP-2 and GPT-4o, and human descriptions typically emphasize semantic content over geometric layout. Nevertheless, our curated question set does include some geometry-oriented items. However, the measured drop in performance is small and offset by significant gains in interpretability and neuroscientific observations. The *Int to CLIP* model serves as an upper bound, revealing that the chosen interpretable space preserves most of the information required for faithful reconstructions.

Category-selective activations align with canonical visual cortex regions and indicate distributed, rather than localized, semantic encoding, with co-activation patterns carrying information beyond

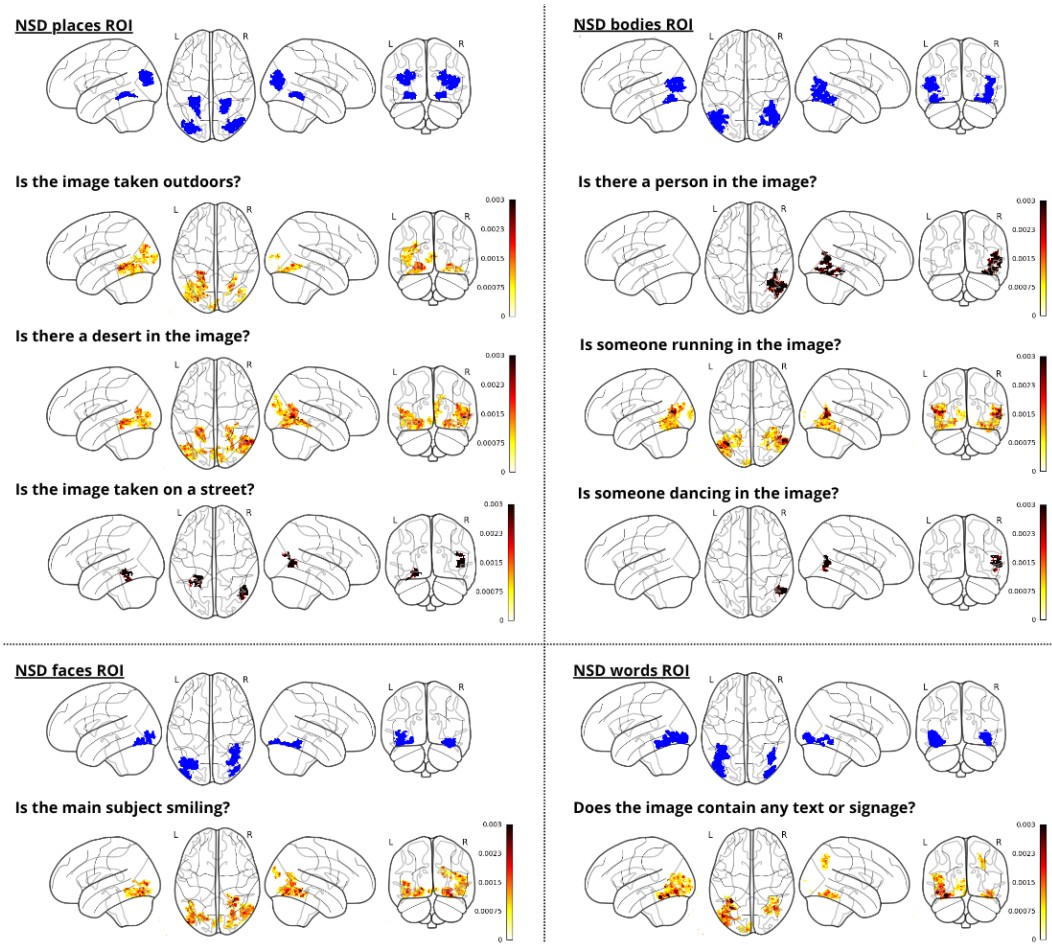

Figure 2: Activation maps in MNI-512 space for places (top left), bodies (top right), faces (bottom left), and words (bottom right). In each section, the first (blue) map is the category-specific reference; subsequent maps show activations from the interpretable matrix $A$. All maps are from subject 01.

voxel location. The semantic space thus bridges neural activity and conceptual representation, consistent with fMRI-scale visual concepts being linearly approximable in vision–language model spaces.

**Limitations and future work**   This work comes with some limitations. First of all, we observe a small performance drop with respect to the non-interpretable baseline. More importantly, interpretability is constrained by the relevance and completeness of the generated question set, and the extension of interpretability for structural information is worth exploring in the future. We also note that the overall quality of the pipeline is limited by the performance of the VQA model used to build the interpretable representations of the images, BLIP-2 in this case. Finally, our study is currently limited to the NSD dataset and to visual decoding tasks. Whether it is possible to generalize to other modalities remains to be demonstrated in future work. Other avenues of future work include enlarging the interpretable space (e.g., hierarchical or structural descriptors), extending to other modalities (such as auditory decoding or language processing), and integrating structural cues such as object layout or scene geometry.

Overall, our results show that interpretability in visual brain decoding can be achieved with minimal cost to reconstruction fidelity, while providing category-selective voxel patterns that overlap with canonical visual cortex hotspots, offering a practical avenue for studying conceptual representations in the brain.

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

# 5 Appendix

## 5.1 Data

In this work we used the Natural Scenes Dataset [2], a benchmark dataset in visual brain decoding from fMRI. NSD data [2] were acquired in high-resolution with ultra-high-field (7T) strength while eight healthy adults subjects performed a continuous recognition task while viewing thousands of images from the COCO dataset. Of a total of eight subjects, we selected only the four subjects who completed all trials (Subjects 01, 02, 05, 07) for easier comparison with other brain decoding works who follow the same protocol. We obtained a training set of 8859 images and 24980 fMRI trials per subject, while the test set included 982 images and 2770 fMRI trials per subject. Because stimuli were presented to subjects up to three times, the corresponding fMRI trials were averaged. The fMRI signal, in 1.8 mm resolution, was masked using the NSDGeneral region-of-interest mask (this includes many areas of the visual cortex), which reduced the spatial dimensionality to approximately 15000 voxels per subject. On the temporal dimensionality, we used NSD-supplied beta weights from a general linear model with fitted hemodynamic response functions; this condensed each voxel's response to a single value per stimulus. Full details of acquisition and preprocessing are available in the original NSD publication [2]. NSD data can be requested at `https://naturalscenesdataset.org/`.

## 5.2 BrainDiffuser

The BrainDiffuser[40] is a well-established brain decoding pipeline consisting of two reconstruction stages that use different latent representations.

**Stage 1: Low-level reconstruction** BrainDiffuser uses VDVAE[11], a hierarchical variational autoencoder with 75 latent layers capturing visual information from coarse to fine. The first 31 layers are selected; during training, their latents $\mathbf{z}$ are extracted from images and concatenated. A ridge regression model is trained to predict these latents from fMRI activations. At test time, predicted latents $\tilde{\mathbf{z}}$ are fed to the pre-trained VDVAE decoder to produce a $64 \times 64$ coarse image, capturing structure but lacking recognisable semantic detail.

**Stage 2: Semantic refinement** The coarse image serves as initialization for Versatile Diffusion (VD)[46], a latent diffusion model that can condition generation on both text and image features. An additional regressor maps fMRI to CLIP-Vision and CLIP-Text embeddings (the latter obtained from COCO captions for each NSD stimulus). At test time, the initial guess is also used: it is encoded by the pretrained AutoKL Encoder of the Versatile Diffusion and noise is added to the obtained latent vector in 37 steps; the so obtained noisy vector is then fed as initialization to the diffusion model and denoised for 37 steps, while conditioning on the predicted CLIP-Text and CLIP-Vision features $\widetilde{\mathbf{h}}$. The result of the diffusion process is then fed to the pretrained AutoKL Decoder of the Versatile Diffusion and the final image reconstruction is obtained. A schematic representation of the BrainDiffuser is available in Figure 3.

## 5.3 Training details

For the "Int to CLIP" model training, we used 5-fold cross-validation to determine the best values of the regularization coefficient $\alpha$ over logarithmically spaced values in the interval $10^{-3} \leq \alpha \leq 10^{4}$.

For the "Brain-to-Int" model, training 5-fold cross-validation was used to determine the optimal values of the regularization coefficient $\alpha$ over logarithmically spaced values in the interval $10^{-6} \leq \alpha \leq 10^{6}$.

All experiments and model training were performed on a server equipped with 8 NVIDIA H100 GPUs, 2 TB of RAM, and 256 CPU threads. The extraction of interpretable latent embeddings with BLIP-2 took approximately 24 hours per subject and the training of the whole interpretable model took less than 30 minutes per subject. The inference time for all models is approximately 3 seconds per decoded image.

## 5.4 Evaluation

To evaluate the performance of the proposed model, we make use of low- and high- level metrics, as done in [39]. Low-level metrics include PixCorr, SSIM, MSE, Cosine Similarity, 2-way accuracy in

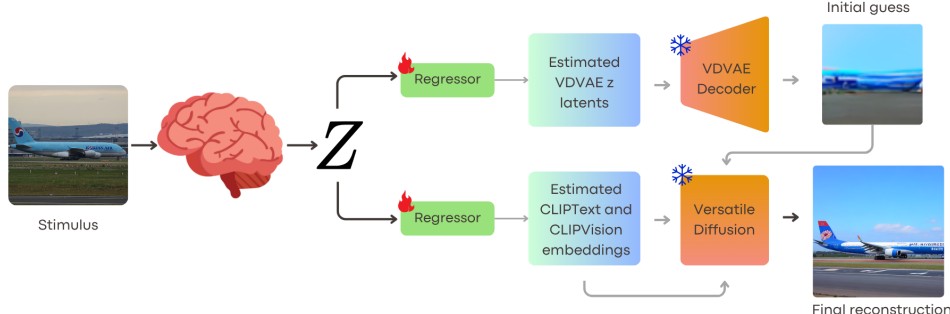

Figure 3: The BrainDiffuser model reconstructs stimuli from fMRI data. A first, structural stream is trained to estimate the representations of a VDVAE while a second, semantic stream estimates CLIP-Text and CLIP-Vision latent features $\widetilde{h}$ from neural data $Z$. Both estimations are done with linear models. The VDVAE latents are fed to the VDVAE decoder to produce a first initial guess of the image, which is given as input to Versatile Diffusion together with the estimated CLIP-Text and CLIP-Vision vectors to obtain the final reconstruction.

the AlexNet latent space (second and fifth layer respectively), while high-level metrics include 2-way accuracy in InceptionV3 and CLIP latent spaces, correlation distance in EfficientNet and SwAV spaces, as well as 50-way-top-1 accuracy using ViT-H/14 (1000 repetitions per image).

## 5.5 Evaluation of interpretable space $\mathcal{L}$

To determine whether the latent space $\mathcal{L}$ retains enough semantic detail to eventually be mapped reliably into the CLIP space by our Interpretable BrainDiffuser, we carried out the following analysis. For every image in the training and test partitions of the NSD we extracted its CLIP-Vision embedding $\mathbf{c} \in$ CLIP and its interpretable embedding $\mathbf{l} \in \mathcal{L}$. We then fit a linear ridge regression model $B : \mathcal{L} \to$ CLIP, selecting the regularization coefficient $\alpha$ on a logarithmic grid from $10^{-6}$ to $10^{6}$ by 5-fold cross-validation. After training, the model projected each test embedding $\mathbf{l}$ into a predicted CLIP vector $\hat{\mathbf{c}} = B\mathbf{l}$. Finally, we performed k-nearest-neighbor retrieval with $k \in \{1, 5, 10, 15\}$. For every test image, we queried the original CLIP index with $\hat{\mathbf{c}}$ and recorded how often the ground-truth image appeared within the top-k returned results. The retrieval accuracy calculated in this way quantifies how faithfully the information in $\mathcal{L}$ can be transferred to CLIP. Table 2 shows the accuracy obtained for the four values of $k$. These results confirm the quality of the designed interpretable representations, further supported by the fact that, even when the ground truth image is absent from the top-$k$ list, the nearest neighbors retrieved by the mapped embeddings typically show a strong similarity with the query. Figure 4 displays a set of representative examples that illustrate this similarity.

Table 2: Top-$k$-Nearest-Neighbors retrieval accuracy achieved after projecting latent embeddings $\mathcal{L}$ into CLIP-Vision space and querying the original CLIP index.

| Metric | Accuracy |
| --- | --- |
| Top-1-NN | 29% |
| Top-5-NN | 59% |
| Top-10-NN | 74% |
| Top-15 | 82% |
| Chance level | 0.1% |

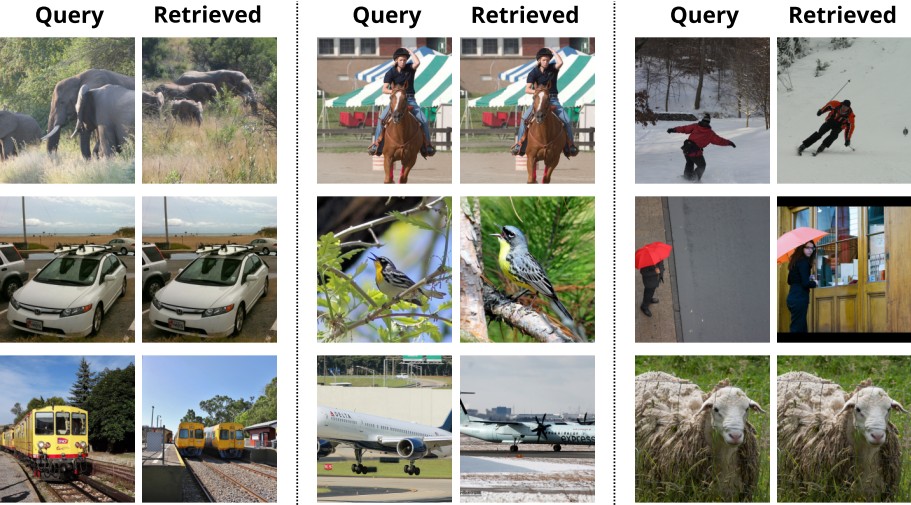

Figure 4: For 9 randomly selected test images (left in each pair), we show the nearest neighbor returned by 1-NN search in the original CLIP index after projecting the latent embedding $\mathcal{L}$ into CLIP space (right in each pair).

## 5.6 Reconstruction examples

Below, we present a series of images reconstructed by our interpretable model, along with their corresponding images generated by the original BrainDiffuser and the *Int to CLIP* model. Images reconstructed by our interpretable model show visual fidelity and similarity to the ground truth images, comparable to the ones achieved by the original, non-interpretable BrainDiffuser. These reconstructed images refer to subjects 1, 2, 5 and 7, respectively.

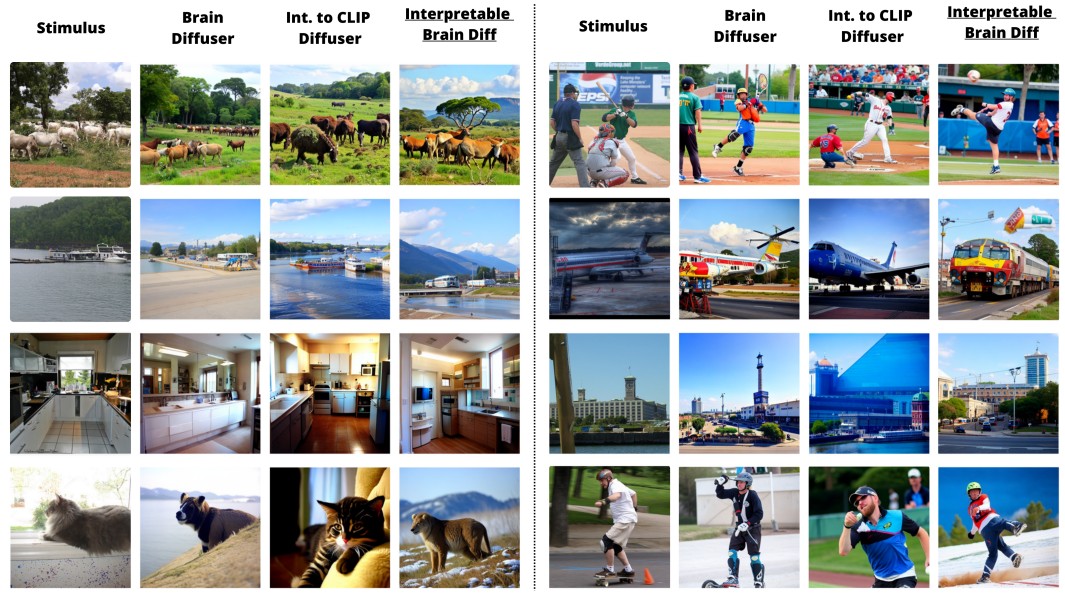

Figure 5: Comparison of our reconstruction results (columns 4, 8) to the original stimuli (columns 1, 5), the images reconstructed by the original BrainDiffuser (columns 2, 6) and those reconstructed by the Int to CLIP model (columns 3, 7). Results presented here refer to subj01.

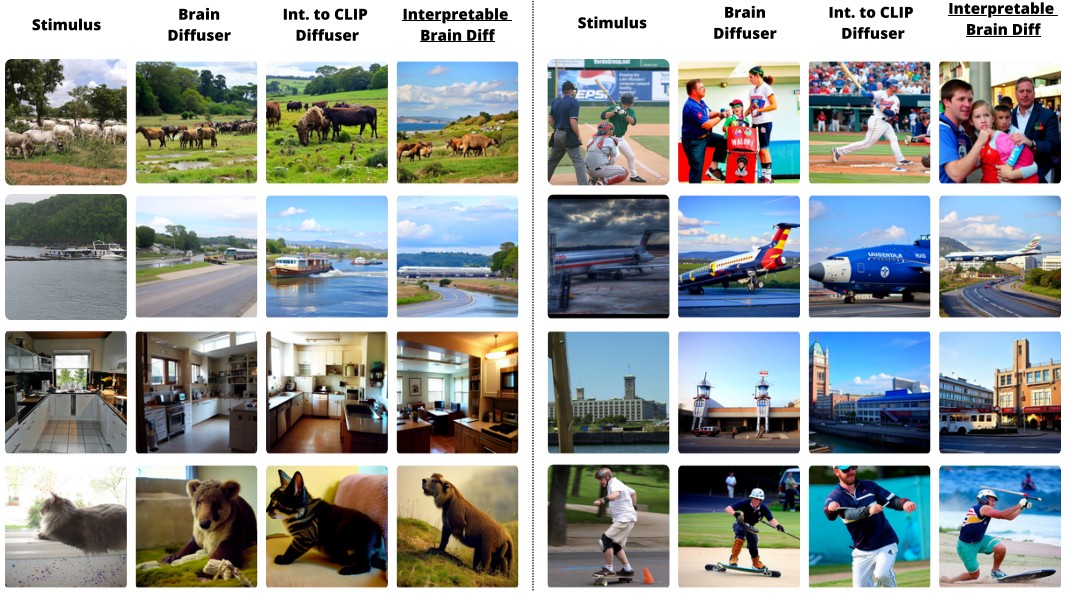

Figure 6: Comparison of our reconstruction results (columns 4, 8) to the original stimuli (columns 1, 5), the images reconstructed by the original BrainDiffuser (columns 2, 6) and those reconstructed by the Int to CLIP model (columns 3, 7). Results presented here refer to subj02.

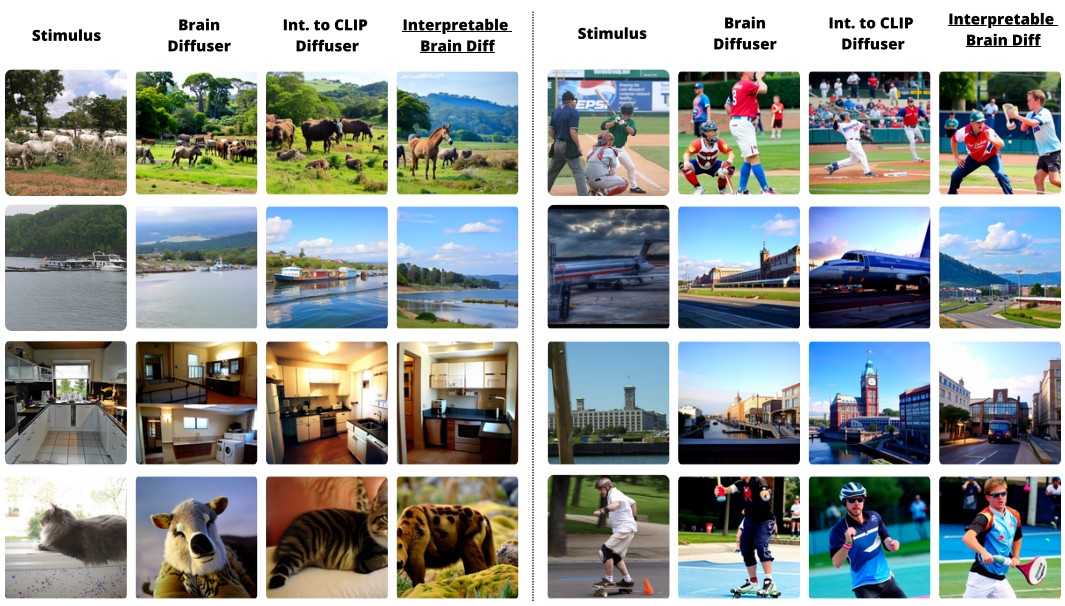

Figure 7: Comparison of our reconstruction results (columns 4, 8) to the original stimuli (columns 1, 5), the images reconstructed by the original BrainDiffuser (columns 2, 6) and those reconstructed by the Int to CLIP model (columns 3, 7). Results presented here refer to subj05.

| Stimulus | Brain Diffuser | Int. to CLIP Diffuser | Interpretable Brain Diff | Stimulus | Brain Diffuser | Int. to CLIP Diffuser | Interpretable Brain Diff |
| --- | --- | --- | --- | --- | --- | --- | --- |

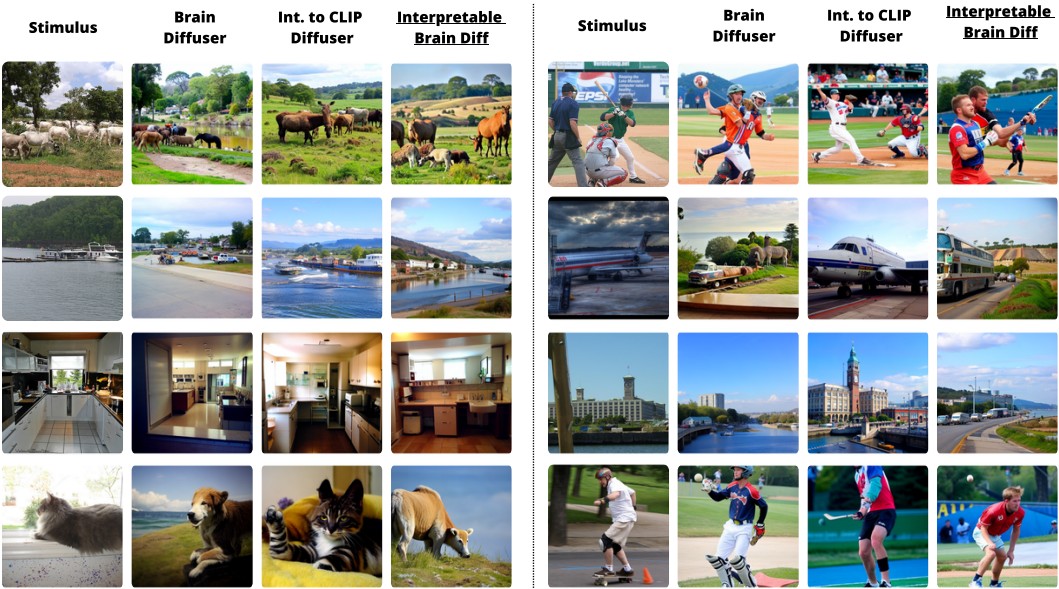

Figure 8: Comparison of our reconstruction results (columns 4, 8) to the original stimuli (columns 1, 5), the images reconstructed by the original BrainDiffuser (columns 2, 6) and those reconstructed by the Int to CLIP model (columns 3, 7). Results presented here refer to subj07.

## 5.7 Brain Region Visualizations

Visualization of brain regions involved most in the estimation of each dimension of $\mathcal{L}$ was achieved by first mapping each Region Of Interest (ROI) to the actual anatomical space of each participant. We then register these coordinates to the MNI-152 standard space, providing a common frame for group-level analyses and comparison with canonical atlases. We compared the resulting maps to visual cortex regions knowingly related to concepts such as bodies, faces, words and places. The analysis was performed in Python using the Nilearn neuroimaging library [1]. For final visualizations, we kept only the top $4\%$ of most influential voxels and discarded any cluster smaller than $100$ voxels to suppress noise.

In this section, we present an extended selection of activation maps derived from the interpretable model for subjects 1, 2, 5, and 7, visualized in MNI-512 space and compared against category-specific reference maps.

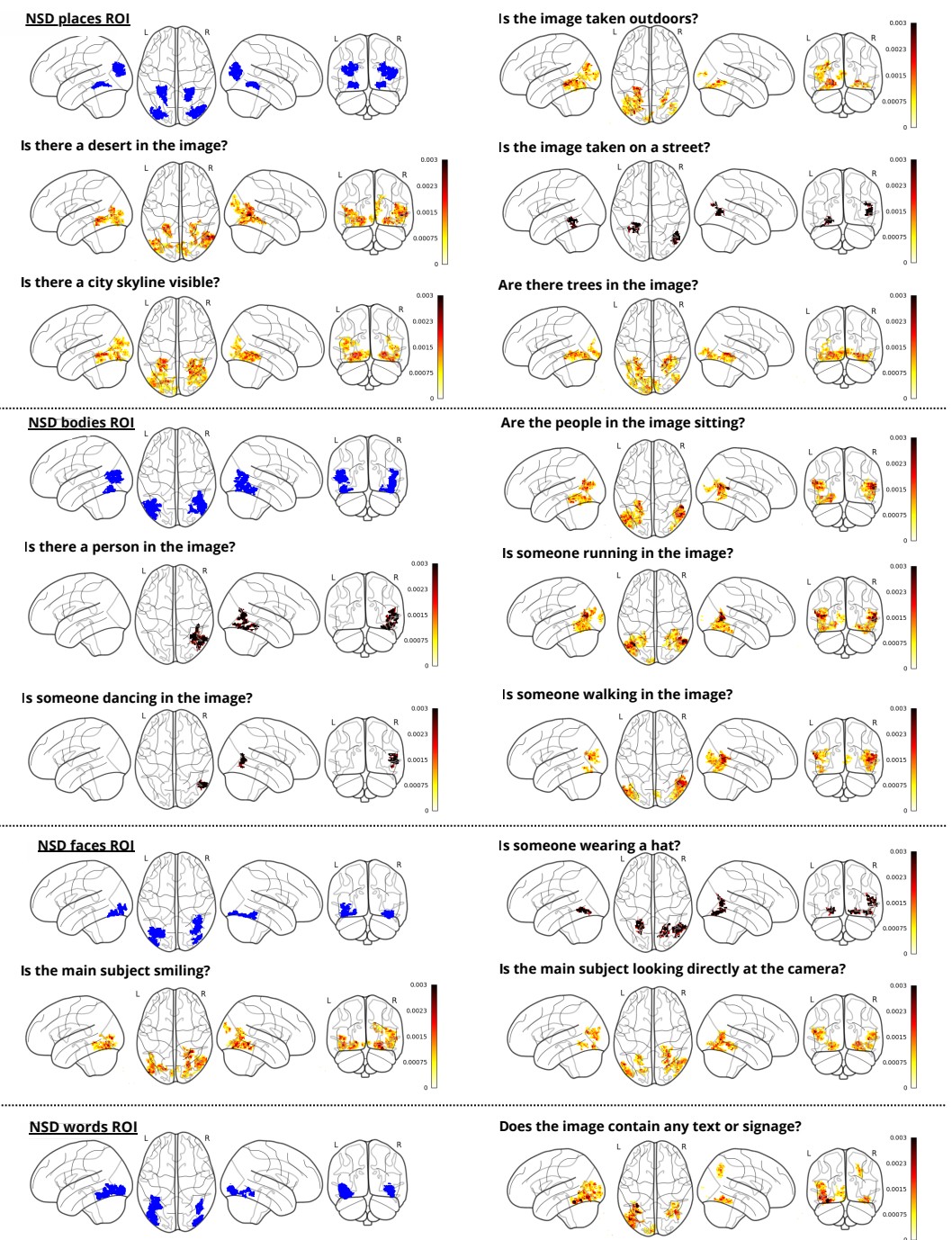

Figure 9: Activation maps for subject 1, projected in MNI-512 space and compared to category-specific reference maps. Sections correspond to places (top left), bodies (top right), faces (bottom left) and words (bottom right). Within each section, the blue map shown first presents the reference for that category. Other images show the activations derived from the interpretable matrix $A$.

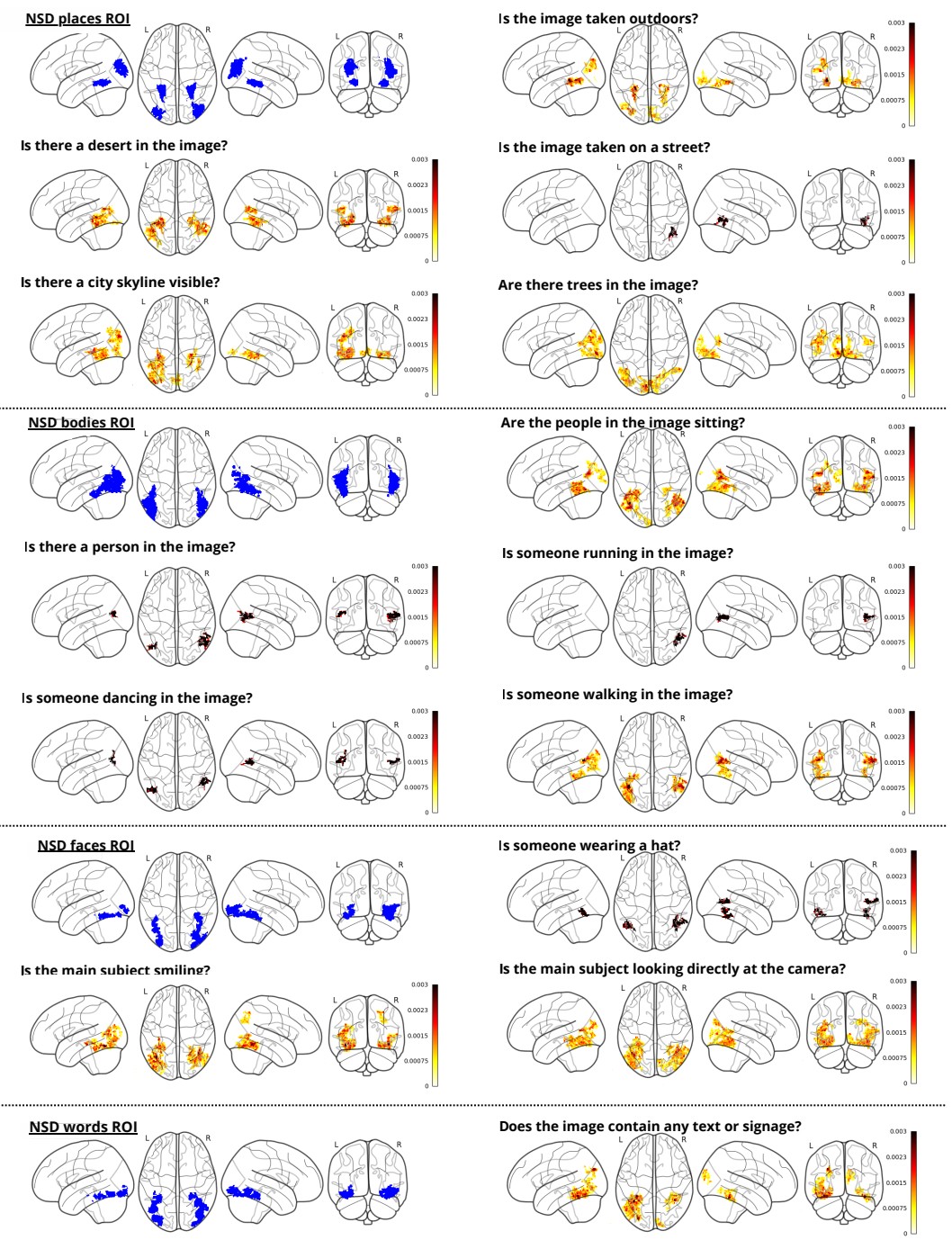

Figure 10: Activation maps for subject 2, projected in MNI-512 space and compared to category-specific reference maps. Sections correspond to places (top left), bodies (top right), faces (bottom left) and words (bottom right). Within each section, the blue map shown first presents the reference for that category. Other images show the activations derived from the interpretable matrix $A$.

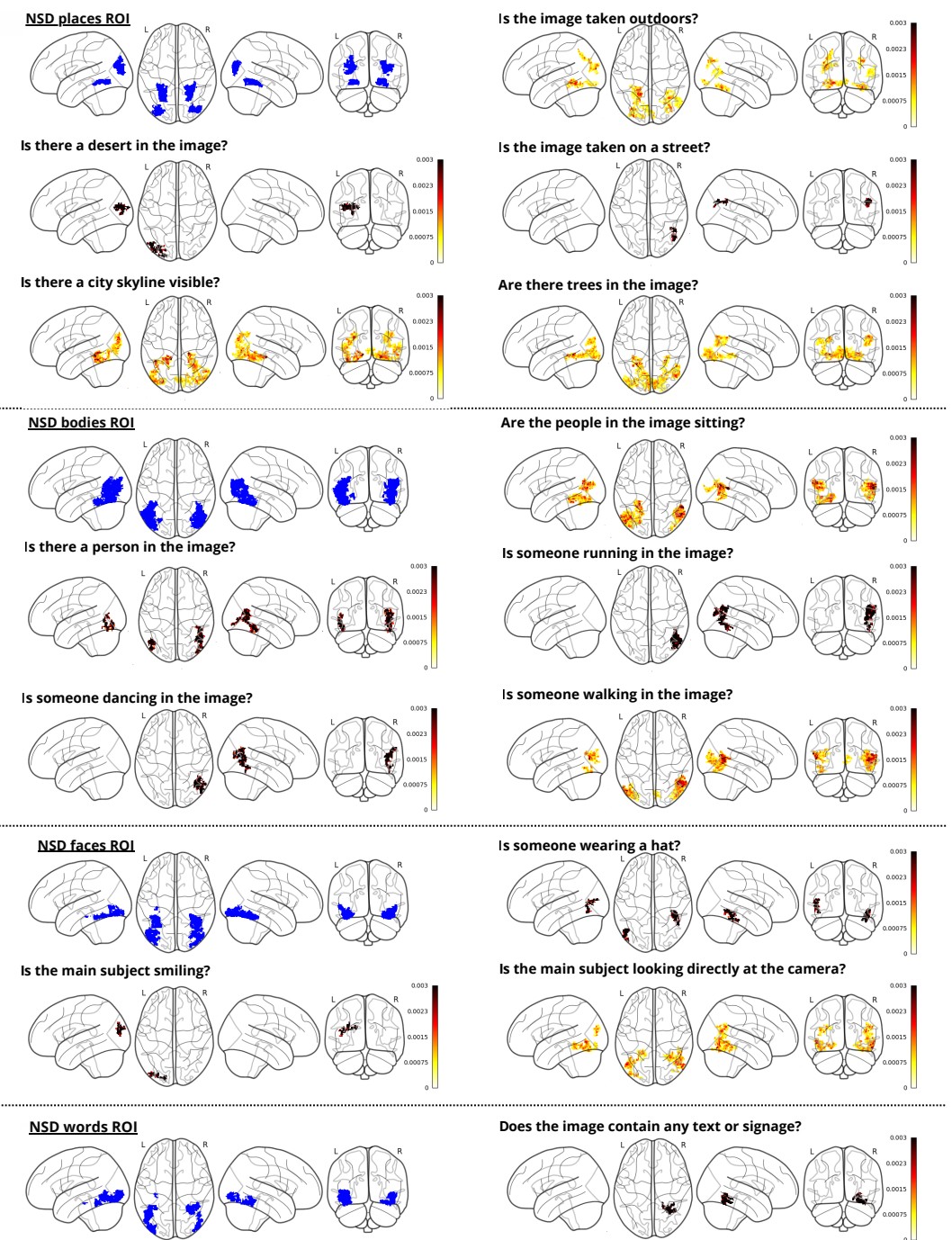

Figure 11: Activation maps, for subject 5, projected in MNI-512 space and compared to category-specific reference maps. Sections correspond to places (top left), bodies (top right), faces (bottom left) and words (bottom right). Within each section, the blue map shown first presents the reference for that category. Other images show the activations derived from the interpretable matrix $A$.

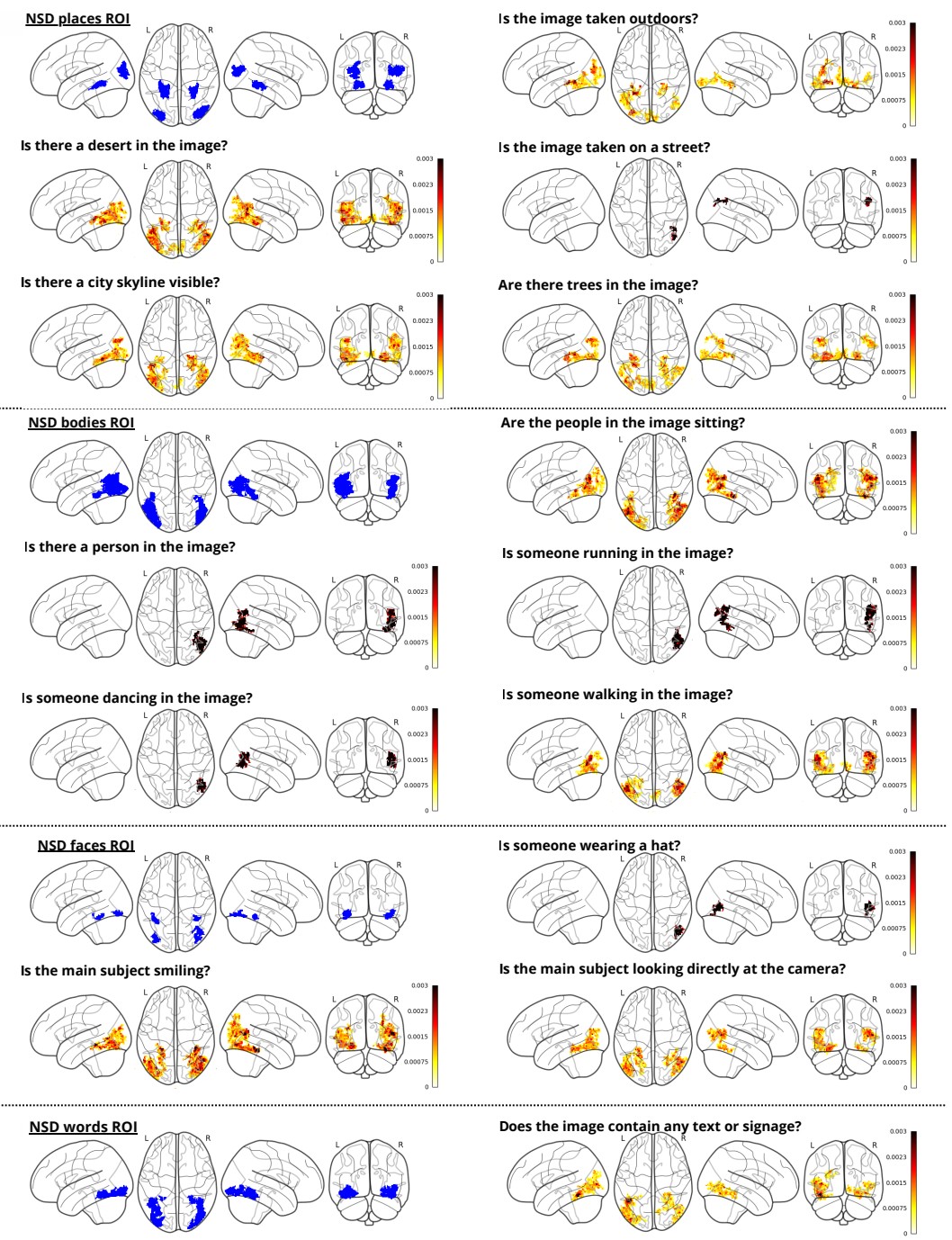

Figure 12: Activation maps for subject 7, projected in MNI-512 space and compared to category-specific reference maps. Sections correspond to places (top left), bodies (top right), faces (bottom left) and words (bottom right). Within each section, the blue map shown first presents the reference for that category. Other images show the activations derived from the interpretable matrix $A$.

## 5.8 Complete list of questions

1. Is there a person in the image?
2. Is there more than one subject?
3. Is the image taken indoors?
4. Is the image taken outdoors?
5. Is the image related to sports?
6. Is the image related to animals?
7. Is the image related to transportation?
8. Does the image contain water bodies like a sea, ocean, or river?
9. Are there any buildings visible in the image?
10. Are there trees in the image?
11. Is the image related to food?
12. Does the image contain any text or signage?
13. Is the image taken during the daytime?
14. Is the image taken at night?
15. Are vehicles present in the image?
16. Are there any children in the image?
17. Are the people in the image standing?
18. Are the people in the image sitting?
19. Is the image related to celebrations or parties?
20. Is the image related to a bathroom or restroom?
21. Is there a surfboard in the image?
22. Is someone riding a wave in the image?
23. Are people sitting on a rooftop?
24. Does the image show a toilet?
25. Are there zebras in the image?
26. Is there a shower visible in the image?
27. Are there animals standing on dirt ground?
28. Is there sand in the image?
29. Is there a mountain in the image?
30. Are there cars parked in the image?
31. Does the image include bicycles?
32. Are there birds in the image?
33. Are there plants or flowers visible?
34. Is someone holding an umbrella?
35. Is someone wearing a hat?
36. Is the image taken on a beach?
37. Are there boats in the image?
38. Is there a bridge in the image?
39. Are there stairs in the image?
40. Is there a window visible in the image?
41. Is there a table in the image?
42. Is there a chair in the image?

43. Is there a bed in the image?
44. Is there a lamp in the image?
45. Are there cups or glasses visible?
46. Are there plates visible?
47. Is there a computer or laptop in the image?
48. Is there a mobile phone visible?
49. Are there books or papers in the image?
50. Are there shelves in the image?
51. Is there a refrigerator in the image?
52. Is there a microwave in the image?
53. Is there a stove or oven visible?
54. Is there a washing machine in the image?
55. Are there curtains in the image?
56. Are there rugs or carpets visible?
57. Are mirrors present in the image?
58. Are there paintings or artworks on the walls?
59. Is there a clock visible in the image?
60. Is there a park in the image?
61. Is the image set in a forest?
62. Is there snow in the image?
63. Is it raining in the image?
64. Is there fog or mist in the image?
65. Are there mountains in the background?
66. Is there a desert in the image?
67. Is there a city skyline visible?
68. Is the image taken on a street?
69. Is there a marketplace in the image?
70. Are there shops or stalls visible?
71. Are there fences or barriers visible?
72. Is the image taken at an amusement park?
73. Are there swings or playground equipment visible?
74. Are there stairs or escalators in the image?
75. Is the image taken in a subway or train station?
76. Are there traffic lights in the image?
77. Is someone swimming in the image?
78. Is someone running in the image?
79. Is someone walking in the image?
80. Is someone cycling in the image?
81. Is someone playing a musical instrument?
82. Is someone reading in the image?
83. Is someone cooking in the image?
84. Is someone eating or drinking?
85. Is someone talking to another person?
86. Is someone taking a photo?

87. Is someone holding an object?
88. Is someone painting or drawing?
89. Is someone driving a vehicle?
90. Is someone fishing in the image?
91. Is someone dancing in the image?
92. Are people wearing jackets or coats?
93. Are people wearing swimsuits?
94. Are people wearing uniforms?
95. Are people wearing traditional clothing?
96. Are hats or caps visible?
97. Is the image symmetrical?
98. Does the image contain bright colors?
99. Are shadows visible in the image?
100. Are reflections visible in the image?
101. Is there smoke or fire in the image?
102. Does the image contain unusual patterns?
103. Are there pets in the image?
104. Are there insects visible in the image?
105. Does the image have signs of damage or destruction?
106. Are there fences or railings visible?
107. Is the main subject a human?
108. Is the main subject an animal?
109. Is the main subject a man?
110. Is the main subject a woman?
111. Is the main subject a child?
112. Is the main subject elderly?
113. Is the main subject a group of people?
114. Is the main subject alone?
115. Is the main subject smiling?
116. Is the main subject interacting with someone?
117. Is the main subject looking directly at the camera?
118. Is the main subject facing away from the camera?
119. Is the main subject partially visible (cropped)?
120. Is the main subject wearing formal clothing?
121. Is the main subject wearing casual clothing?
122. Is the main subject carrying an object?
123. Is the main subject holding a tool?
124. Is the main subject using electronic devices?
125. Is the main subject sitting on the ground?
126. Is the main subject climbing?
127. Is the animal a mammal?
128. Is the animal a bird?
129. Is the animal a reptile?
130. Is the animal an amphibian?

131. Is the animal a fish?
132. Is the animal a pet?
133. Is the animal a farm animal?
134. Is the animal a wild animal?
135. Is the animal flying?
136. Is the animal swimming?
137. Is the animal eating?
138. Is the animal drinking water?
139. Is the animal alone?
140. Are there multiple animals?
141. Are the animals interacting with each other?
142. Is the animal domesticated?
143. Is the main subject an object?
144. Is the object made of wood?
145. Is the object made of metal?
146. Is the object made of plastic?
147. Is the object broken?
148. Is the object old or vintage?
149. Is the object modern?
150. Is the object electronic?
151. Is the object artistic or decorative?
152. Is the object used for work or utility?
153. Is the main subject in the center of the image?
154. Is the main subject on the left side?
155. Is the main subject on the right side?
156. Is the main subject near the top?
157. Is the main subject near the bottom?
158. Is the main subject partially out of the frame?
159. Is there a background behind the main subject?
160. Is the main subject framed by other objects?
161. Is the main subject closer to the foreground?
162. Is the main subject farther in the background?
163. Does the image appear staged?
164. Does the image look candid or natural?
165. Is there movement captured in the image?
166. Is the image static?
167. Does the image look artistic or abstract?
168. Is the setting rural?
169. Is the setting urban?
170. Is the setting domestic?
171. Is the setting industrial?
172. Is the setting natural?
173. Is it sunny in the image?
174. Is it cloudy in the image?

175. Is there rain in the image?
176. Is it snowing in the image?
177. Are there visible shadows in the image?
178. Does the image depict sunset or sunrise?
179. Is the image taken during golden hour?
180. Is someone dancing?
181. Is someone playing sports?
182. Is someone cooking?
183. Is someone cleaning?
184. Is someone fixing or repairing something?
185. Is someone driving?
186. Is someone hiking?
187. Is someone fishing?
188. Is the image black and white?
189. Is the image edited or filtered?
190. Does the image use high contrast?
191. Is the image blurred?
192. Does the image have reflections?
193. Is the subject interacting with animals?
194. Is the subject interacting with machines?
195. Is the subject interacting with nature?
196. Is the subject interacting with others?
197. Is the subject interacting with water?
198. Is the subject wearing sunglasses?
199. Is the subject wearing a hat?
200. Is the subject wearing a uniform?
201. Is the subject wearing shoes?
202. Is the subject barefoot?
203. Is there a car visible?
204. Is there a bicycle visible?
205. Is there a bus visible?
206. Is there a train visible?
207. Is there an airplane visible?
208. Is there a boat visible?
209. Is there a motorcycle visible?
210. Is there symmetry in the image?
211. Is there repetition or patterns in the image?
212. Does the image depict destruction or ruins?
213. Are there visible tools or instruments?
214. Is there a stage or performance area visible?

