# OpenReview forum: "One Question at a Time: A Semantic Bottleneck for Interpretable Visual Brain Decoding from fMRI"
_NeurIPS.cc/2025/Workshop/UniReps — UniReps2025_

### Official Review · Reviewer_QnEu · 2025-09-05

**Confidence:** 3

**Review:**

## Summary
 - This paper proposes an interpretable brain decoding framework for fMRI-to-image reconstruction.

## Strengths
 - Novel idea of inserting a semantic bottleneck into an existing high-performing pipeline.
 - Clear motivation: addressing interpretability in brain decoding, a recognized limitation.
 - Solid methodology: factorization into A (voxel-to-concept) and B (concept-to-CLIP) is elegant.
 - Empirical results show only minor performance degradation, with strong neuroscientific interpretability.
 - Paper is clearly written and well-structured.

## Weaknesses
 - Performance Eval : More discussion of low-level metrics (SSIM, PixCorr) would strengthen the case.
 - The semantic question set (214 binary dimensions) is somewhat arbitrary; unclear how robust the method is to different question sets.
 - Heavy reliance on BLIP2 for question answering introduces a dependency that may bias embeddings; not fully explored.
 - Experiments are limited to NSD dataset; generalization to other datasets or modalities (e.g., MEG, EEG) remains untested.

## Questions for Authors
 - How sensitive are the results to the choice or number of semantic questions?
 - Could the semantic bottleneck be hierarchical (e.g., coarse-to-fine categories)?
 - How does the model handle noisy or ambiguous BLIP2 answers in constructing embeddings?
 - Do you expect this approach to transfer to other tasks (e.g., language decoding) or modalities (EEG/MEG)?

## Recommendation
 Good paper. Accept

**Score:**

4

**Topic Fit:**

2

---

### Official Review · Reviewer_RJan · 2025-09-14
**Limited Contribution and Shallow Neuroscientific Insight in Semantic Bottleneck Extension of BrainDiffuser**

**Confidence:** 5

**Review:**

The paper proposes a semantic bottleneck module integrated into the existing BrainDiffuser pipeline for visual image reconstruction from brain activity. In the abstract, the authors claim that their approach has a lower decoding performance compared to the original BrainDiffuser while providing improved interpretability and neuroscientific insight. However, the paper’s contributions are limited, and several key issues reduce its impact.

Main Concerns:

1. Lack of Contribution and Performance Improvement

The method is built upon the well-established BrainDiffuser framework. The only modification is the inclusion of a semantic feature bottleneck during the image reconstruction process. However, the results do not demonstrate any clear improvement in reconstruction accuracy over the original BrainDiffuser. In fact, performance appears slightly lower, as acknowledged by the authors themselves.

Moreover, the evaluation lacks comprehensive comparison with other advanced models in the field, such as MinDEye 1, MinDEye 2, and Dream etc. Without broader benchmarking, it is difficult to assess the actual value added by the proposed semantic bottleneck.

2. Limited and Superficial Neuroscientific Insight

The authors claim improved neuroscientific interpretability as the main contribution, yet the paper falls short in delivering meaningful insight. In the results section, the discussion of neural findings is limited to superficial observations, such as differences in brain activation for face/body/place prompts.

There is no in-depth analysis of how semantic prompts relate to known visual regions like PPA, OPA, RSC, or LOC, nor any exploration of functional connectivity or theoretical rationale behind the observed neural activity. The neuroscience discussion lacks rigor and fails to support the claimed insight, which is problematic since this is positioned as the paper’s main novelty over BrainDiffuser.

Minor Concerns:

1. Reference Formatting Issues

The reference list in the main text is disorganized. For example, In introduction ...[47, 27, 26, 29, 25, 13, 28, 35, 39, 19, 21, 5, 9, 10, 15, 45] is poorly formatted and not in order, which affects readability and professionalism.

2. Inconsistent Dataset Citation

The NSD dataset is cited inconsistently. In the Methods section, it's cited as [2], while in the Appendix 5.1, it's cited as [3]. The authors should ensure all references are cited consistently and correctly throughout the manuscript.

**Score:**

1

**Topic Fit:**

2

---

### Official Review · Reviewer_Abe6 · 2025-09-15
**BLIP Derived Latent Space for Brain Diffusion: Enhancing Neuroscientific Insight with Minimal Performance Loss**

**Confidence:** 3

**Review:**

The authors propose a new decoding mechanism that extends the standard Brain Diffusor approach. Instead of mapping brain activity directly into the high-dimensional, opaque CLIP vision language embedding space, they construct a 214 dimensional interpretable latent space derived from binary question answer pairs generated by BLIP. They then factorize the weight matrix so that brain responses are first projected into this latent space, and separate regressors subsequently map it to CLIP embeddings. This design addresses the limited interpretability of the direct brain-to-CLIP mapping, enabling neuroscientific insights into the brain’s functional organization. Although this method incurs a modest drop in reconstruction quality compared with the original Brain Diffusor, its key strength lies in providing a more interpretable framework for neuroscientific analysis.

Cons/Suggestions -

While the proposed latent space is an interesting step toward interpretability, the neuroscientific insights it offers remain underexplored, admittedly a challenge within the constraints of a four-page paper. For example, it would be valuable to quantify whether the learned latent matrix meaningfully groups voxels according to ROI-specific functionality. Could we define a metric that, for instance, shows that n percent of voxels in the fusiform face area (FFA) contribute most strongly to face-related images compared with voxels in other regions?

Beyond such region-specific contributions, what additional insights emerge? Established findings show that higher visual areas encode more semantic information, whereas lower areas are more sensitive to spatial details. Given that early visual areas (e.g., V1–V3) are primarily driven by low-level spatial features, it would be surprising to find voxels there that are strongly selective for high-level semantic concepts. What, if anything, does the proposed latent space reveal about these regions?

**Score:**

4

**Topic Fit:**

3

---

### Official Review · Reviewer_SUNG · 2025-09-15
**Adding a “semantic bottleneck” to brain-to-image decoding, mapping fMRI activity into a 214-dimensional space of yes/no visual questions**

**Confidence:** 4

**Review:**

This paper adds a “semantic bottleneck” to brain-to-image decoding, mapping fMRI activity into a 214-dimensional space of yes/no visual questions before projecting into CLIP embeddings for reconstruction. The approach keeps reconstruction accuracy close to BrainDiffuser while offering interpretable voxel-to-concept maps. These maps align with known visual regions (faces, places, bodies), probably suggesting neuroscientific validity.

Things I liked:
1- Introduces a semantic bottleneck that links fMRI signals to human-understandable concepts, making voxel-level analysis possible.
2- Produces voxel-to-concept maps that align with canonical visual regions (FFA, PPA, EBA), supporting the biological plausibility of the method.


Negative points that I am concerned about:

1- The 214 GPT-generated binary questions define the entire semantic space. There is no analysis of coverage, redundancy, or whether missing concepts limit reconstruction fidelity or neuroscientific validity.
2- The quality of the interpretable embeddings depends entirely on BLIP2’s accuracy in answering binary questions. Mislabels could propagate noise downstream. I am suggesting that human validation or error sensitivity analysis is needed.
3- While voxel activation maps are visually compelling, there’s no statistical measure of overlap with known functional ROIs or quantification of semantic alignment. This weakens the interpretability claim a lot.
4- Experiments are limited to NSD (8 subjects, 7T fMRI). Generalization to different datasets, scanners, or naturalistic settings remains untested.
5- Although Fig. 2 provides valuable illustrative evidence that the semantic bottleneck aligns voxel activations with canonical visual regions, its validity is limited because: 1) no quantitative overlap metrics (permutation tests), 2) results shown only for one subject, lacking cross-subject consistency.

**Score:**

3

**Topic Fit:**

3